# Domain Adaptation for MRI Organ Segmentation using Reverse Classification Accuracy

**Vanya V. Valindria**[1]**, Ioannis Lavdas**[2]**, Wenjia Bai**[1]**, Konstantinos Kamnitsas**[1]**,
Eric O. Aboagye**[2]**, Andrea G. Rockall**[3]**, Daniel Rueckert**[1]**, and Ben Glocker**[1]

[1]Biomedical Image Analysis Group, Imperial College London, UK
[2]Comprehensive Cancer Imaging Centre, Hammersmith Hospital, Imperial College London, UK
[3]The Royal Marsden NHS Foundation Trust, London, UK
`v.valindria15@imperial.ac.uk`

## Abstract

The variations in multi-center data in medical imaging studies have brought the necessity of domain adaptation. Despite the advancement of machine learning in automatic segmentation, performance often degrades when algorithms are applied on new data acquired from different scanners or sequences than the training data. Manual annotation is costly and time consuming if it has to be carried out for every new target domain. In this work, we investigate automatic selection of suitable subjects to be annotated for supervised domain adaptation using the concept of reverse classification accuracy (RCA). RCA predicts the performance of a trained model on data from the new domain and different strategies of selecting subjects to be included in the adaptation via transfer learning are evaluated. We perform experiments on a two-center MR database for the task of organ segmentation. We show that subject selection via RCA can reduce the burden of annotation of new data for the target domain.

## 1 Introduction

Machine learning has led to significant advances in medical imaging, particularly with big improvements in medical image segmentation. Performance, however, depends on the availability of sufficient amounts of labeled samples for supervised learning, and also whether the test data is coming from the same domain as the training data. In clinical practice, the source domain (S) on which the classifier is trained might be different from the target domain (T) with clinical data. The images from these domains are samples from different appearance distributions. The mismatch of distributions is caused by various factors such as the use of different scanners, types of sequences, or biases in patient cohorts - often causing a trained algorithm to perform poorly on new data. In a scenario where the tasks are the same, but the source and target domains are different, domain adaptation is usually performed to address the domain disparity problem [7].

Domain adaptation can be categorized into three settings, supervised, semi-supervised, and unsupervised [8, 4]. Our work focuses on supervised domain adaptation methods, which uses labeled data from the target domain. In the context of convolutional neural networks (CNNs), supervised domain adaptation can be approached by training from scratch or fine-tuning a network pre-trained on the source domain [9].

An approached called DALSA [3] explored a supervised domain adaptation by introducing a weighting scheme in Random Forests and SVMs for segmentation. This approach preserves the segmentation quality even though only sparse annotated data is used. Another approach tackles the segmentation difficulty of images from different scanners and imaging protocols by weighting different SVM-based

classifiers for transfer learning [13]. More recent work has tried to explore transfer learning in CNNs in medical imaging [10, 11], which confirmed the potential of fine-tuned and fully trained CNNs.

However, the importance of subject selection in transfer learning has not been widely studied yet. Most works [1, 8, 14] attempt to integrate active learning with domain adaptation. Intuitively, active learning is chosen since it develops a criterion to determine the "value" of a candidate for annotation.

In active domain adaptation, the classifier selects among the limited labels on target data [8] by combining hybrid oracles. However, obtaining the oracles is costly. Instead of selecting instances to be added to the training set, [1] selects a bag of instances by self-training. But, this approach is more effective when the source and target domains differ substantially. An appealing work is a recent application of active learning in domain adaptation for biomedical images [14]. Active learning chooses the candidates with higher entropy and higher diversity, which are expected to improve the current performance. In [14], a pre-trained CNN is further fine-tuned continuously by incorporating newly annotated samples in each iteration to enhance the performance incrementally. Although they can reduce the annotation cost, the iterative scheme can be time consuming, especially if applied to volumetric data. Previous work on subject selection with active learning [1, 8, 14] requires iterations and only deals with classification of 2D images.

Instead of using iterative active learning, we propose a framework to select the "most valuable" samples from the unlabeled target domain to be annotated - using reverse classification accuracy (RCA) [12]. We address the question whether RCA can be employed to select few subjects to reduce the cost of annotation in supervised domain adaptation. To answer this question, we systematically conducted several experiments. Our contributions are: (1) demonstrating the effective use of RCA as a selector for $n$-subjects in target domain to be incorporated into the training dataset, (2) we compare different strategies for supervised domain adaptation with the RCA selection, (3) we study how the training size and the combination of target samples affects segmentation performance.

## 2 Materials and Methods

### 2.1 Datasets

*Source domain (S):* The dataset is obtained from our MALIBO study (MAchine Learning In whole Body Oncology) and includes abdominal T1-weighted MR Dixon images of 35 healthy subjects. We consider this dataset as the source domain used to train the initial classifier in a supervised manner as manual organ annotations are available for all subjects. The images have size of $(256 \times 208 \times 202)$ and resolution $(1.64 \times 1.64 \times 5)$ mm.

*Target domain (T):* Data for the target domain is obtained from the UK Biobank (Application 12579). We use 45 subjects with manually annotated T1-weighted Dixon MR images which have been acquired with a similar protocol as the MALIBO data. The main obvious differences are the image size $(224 \times 168 \times 366)$ and resolution $(2.23 \times 2.23 \times 3)$ mm.

Source and target dataset are acquired at different centers. UK Biobank data is acquired with a Siemens 1.5T MAGNETOM Aera scanner while in MALIBO a Siemens 1.5T MAGNETOM Avanto was used. For our study, we use the T1-weighted in-phase images from the Dixon protocol. We focus on organ segmentation within whole-body scans. As a pre-processing, image intensities in both datasets are normalized to zero-mean and unit-variance. Images are resampled to the same size and physical resolution. Visual examples from the source and target database are depicted in Figure 1. Despite the similarity of the scanning protocol and same scanner manufacturer, the drop in segmentation accuracy when applying a model trained on MALIBO and tested on UK Biobank data is striking, as we will show in our experiments. The images seem to encode a significant bias in their appearance which is not obvious upon visual inspection.

### 2.2 Reverse Classification Accuracy

Reverse classification accuracy (RCA) is a recent method for predicting the segmentation accuracy in the absence of ground truth [12]. RCA can accurately predict the quality of a segmentation on a case-by-case basis without the need for labeled testing set. Here we apply RCA with the single-atlas registration classifier as described in [12]. The test image together with its predicted segmentation is registered to a set of reference images such that the predicted segmentation can be

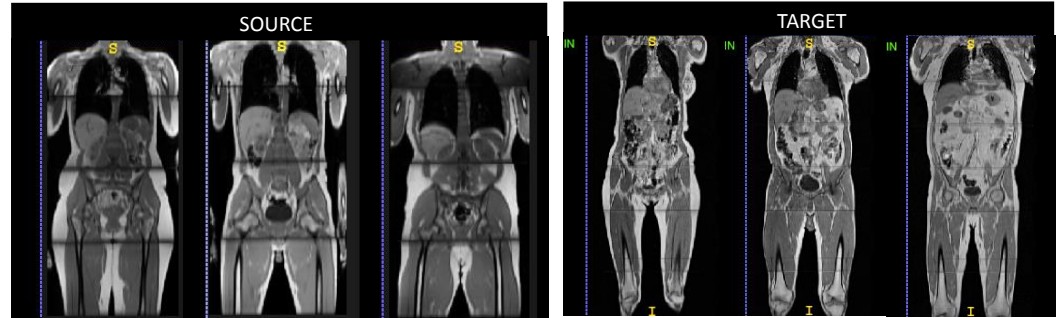

Figure 1: Examples of whole-body scans from source (MALIBO) and target (UK Biobank) database.

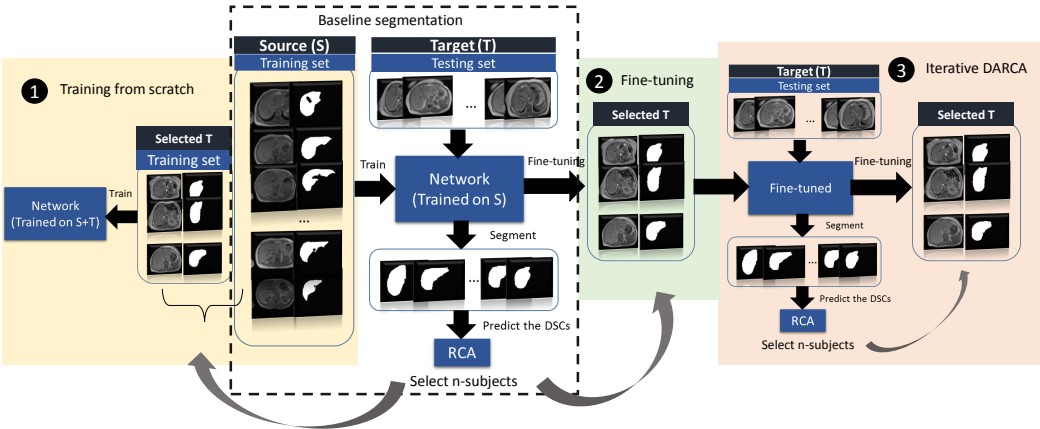

Figure 2: Overview of the strategies we tried for $n$-subject selection in DARCA: 1. Training from scratch. 2. Fine-tuning. 3. Iterative DARCA

quantitatively compared to the manual segmentations of the reference images by computing Dice similarity coefficients (DSC). It is expected that the maximum DSC score over all reference images correlates well with the real DSC. For more details on the RCA framework, see [12].

RCA acts as a selector for picking up subjects with high and low confidence in segmentation accuracy. Our hypothesis is that transfer learning with specific $n$-subject selection is better than picking-up random subjects from the target domain, and thus fewer manually labelled subjects are needed from the target domain. In the following, we call this 'domain adaption using RCA', or DARCA.

### 2.3 Supervised Domain Adaptation

In our experiments, we employ DeepMedic[1] [5] as the base network for 3D organ segmentation. We use the default 11-layers deep, multi-scale, parallel convolutional pathways architecture.

The main approaches of supervised domain adaptation with CNNs are either training from scratch or fine-tuning [11]. Using RCA as a selector, $n$-subjects from the target domain are added to the training data. With the new training data containing source and $n$-target (S+T) subjects, we train the network from scratch. Meanwhile, we can transfer the parameters from a pre-trained network and fine-tune on another database. To build the model, we fine-tune the pre-trained network (from S-only training), with the $n$-target subjects selected by RCA. Based on the results in [2], fine-tuning the last layer achieves the best performance compared to using more convolutional layers for fine-tuning. We fine-tuned only the last layer of the pre-trained networks and used the same optimization but with fewer epochs. Based on our experiments fine-tuning all the layers is shown to have lower accuracy and more training time is needed.

---

[1]Source code available at https://github.com/Kamnitsask/deepmedic

| Training | DSC (mean (stdv)) |
|---|---|
| Train on S (baseline) | 0.639 (0.149) |
| Train on T (upper-bound) | 0.873 (0.046) |

Table 1: Baseline and upper-bound accuracies. The upper-bound gives the highest performance when the network is trained and tested on one domain (target dataset). The performance drops significantly when training and testing data are from different domains (baseline).

## 3    Experiments and Results

We present results for using different strategies to investigate the effect of RCA-based subject selection for domain adaptation, as shown in Figure 2. We use 3-fold cross-validation with the same random splits in all experiments. As the baseline, we trained the network with all S data and tested it to segment the T images. We predict the DSCs of all target segmentations using RCA. After we sort their DSCs (lowest to highest confidence), we select $n$-subjects from T domain to be included with their corresponding manual annotations in the training set mimicking an active learning approach.

### 3.1    Training from scratch

We set our baseline as the segmentation of T with an S-only trained network, whereas the upper-bound is the segmentation of T with T-only trained network on liver segmentation, as shown in Table 1.

Figure 2-1 shows that RCA selects $n$-subjects from T domain, to be manually labeled and incorporated into training dataset. For this experiment, we compared best-/worst-5 subjects selected by RCA and the real best-/worst-5 (real DSCs from the target ground-truth). Besides, we also run random-5 subject selection (repeated with three different random combination and taking the average) to be trained from scratch.

In the training-from-scratch strategy, we train S+T data simultaneously with the same optimization, update-rule, number of epochs, loss function, and regularization techniques as in the baseline. From Table 2 column one, we can see that picking up 5 subjects from target domain has already improved the accuracy, compared to the baseline. Table 2 shows that the segmentation accuracy using the best-5 or worst-5 subject selection outperformed the random selection. Moreover, RCA selection gives a relatively similar segmentation accuracy to the selection using real DSC.

As we increase the number of annotated target images to be incorporated in training from scratch, the accuracy does improve, as shown in Table 3. From Table 2, T segmentation with lowest confidence also gives a good contribution to the current CNN, therefore we combine 5 target subjects with highest confidence and 5 subjects with lowest confidence into the training. The result shows that this combination achieves similar accuracy to the one that incorporated all annotated T subjects. Hence, we reduce the "less valuable" samples for training by RCA selection. Unfortunately, training from scratch requires more time since the networks must learn from the beginning.

### 3.2    Fine-tuning a Pre-trained Network

In Figure 2-2, we fine-tune the pre-trained network (from S), with $n$-subjects selected by RCA. Fine-tuning requires less time than training from scratch. Here, we fine-tune with three different selections (random, real DSC, and RCA) at different set size (2, 5, 10, 15, and all of the T data). Table 3 shows the results when we fine-tune using all of the T data which is very similar (DSC: 0.830) to when we train from scratch (DSC: 0.831).

However, RCA seems worse to predict the real segmentation accuracy of fine-tuning. There is a gap between best-/worst-5 real selection and best-/worst-5 RCA accuracy (Table 2). One of the reasons could be due to the under-estimation of RCA prediction on the baseline segmentation accuracies on T, with 0.88 correlation and 0.15 MAE. As explained in the experiments of multiple-organs segmentation [12], the RCA prediction for organs with the real DSCs between (0.6 - 0.8) is not as accurate as in organs with real DSCs above 0.8. In our case, the average real DSCs of baseline liver segmentation is 0.639, but RCA under-estimated the mean of predicted DSCs to be 0.497.

| Strategies of sample selection | Training from scratch | Fine tuning | Iterative |
|---|---|---|---|
| Baseline | 0.639 (0.149) | 0.639 (0.149) | 0.639 (0.149) |
| All T | 0.831 (0.074) | 0.830 (0.066) | n/a |
| Random 5 | 0.720 (0.103) | 0.710 (0.172) | n/a |
| Worst 5 (real) | 0.797 (0.051) | 0.619 (0.256) | n/a |
| Worst 5 (RCA) | 0.799 (0.048) | 0.771 (0.156) | **0.828** (0.072) |
| Best 5 (real) | 0.747 (0.152) | 0.723 (0.173) | n/a |
| Best 5 (RCA) | 0.755 (0.148) | 0.687 (0.191) | 0.777 (0.107) |
| Best 5 and Worst 5 (real) | 0.823 (0.058) | **0.842** (0.050) | n/a |
| Best 5 and Worst 5 (RCA) | **0.831** (0.063) | 0.835 (0.065) | n/a |

Table 2: Strategies of DARCA on liver segmentation: Training from scratch, fine-tuning, and iterative scheme (only for 2nd iteration with RCA selection) with different choice of subject selection.

As we did in Section 3.1, we also combined best-5 and worst-5 in T domain to be annotated and incorporated in fine-tuning. This 10 combined subjects give much better accuracy than when we choose only the best-10 subjects (Table 3). Similarly, picking up the worst sample in increasing number, results in lower accuracy than the 'best 5 and worst 5'. Best-5 and worst-5 annotated samples with real selection shows the best results (DSC: 0.842), and RCA selection also gives similar results (DSC: 0.835). Additionally, this best-5 and worst-5 combination (real and RCA selection) performs better than when we use all of the annotated subjects from T (DSC: 0.830). Hence, we cut the cost of annotation by 67%, using only 10 selected subjects instead of 30 subjects and achieve a higher accuracy.

### 3.2.1 Pseudo-labels for Fine-tuning

In this experiment we investigate the use of pseudo ground-truth labels in a semi-supervised way. In Section 3.2, we incorporated the $n$-subjects by fine-tuning with their ground-truth. What if, instead of using the real annotations, we use the predicted labels as pseudo ground-truth, which are the baseline segmentation results - for training. In the previous work by [6], pseudo-labels are used for semi-supervised learning with pre-trained and fine-tuning scheme. Pseudo-labels are defined as labels that have maximum predicted probability and seen as equivalent to entropy regularization, which encourages low density separation between classes.

However, from Table 3, it is clear that using pseudo-labels cannot improve the segmentation performance on the target domain. Fine-tuning with all of the pseudo-labelled subjects in T gives the worst result amongst all. The noisy labels negatively impact the segmentation performance. Hence, training using pseudo-labels seems not suitable for domain adaptation in our application, since it assumes the baseline classifier to be of good quality, while by default it should be considered to be severely suboptimal.

### 3.3 Iterative DARCA

Different from the previous strategies, here, we wish to mimic the active learning domain adaptation [8], where at each iteration, RCA chooses $n$-subjects from the target domain to fine-tune the baseline networks (see Figure 2-3). At the first iteration, we fine-tune the baseline network with best-5 subjects selected by RCA. This new network is used to segment the test images from target domain for which accuracy is again predicted using RCA. At the second iteration, we fine-tune the network again with the the best-5 and worst-5 selected subjects.

The results in Table 3.2 show that the second iteration with worst-5 subjects gives higher accuracies than fine-tuning with best-5 RCA. The combination of best-5 (1st iteration), and worst-5 (2nd iteration) by RCA performs almost the same (DSC: 0.828) as fine-tuning using all of the labeled target data (DSC: 0.830). Additionally, the 2nd iteration with worst-5 RCA selection generally improves the 1st iteration (by best-5 RCA selection) accuracies. Hence, with less labeled data we can save time with similar results.

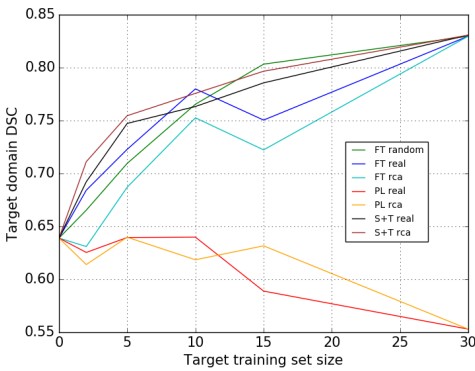

Figure 3: Plot for different $n$-selection training size in different strategies: Fine-tuning (FT), fine-tuning with pseudo-labels (PL), and training from scratch (S+T). Similar trends are shown between real and RCA selection on FT and S+T with different size.

| Strategies | 0 | 2 | 5 | 10 | 15 | 30 (all) |
|---|---|---|---|---|---|---|
| FT random-n | 0.639 (0.149) | 0.665 (0.245) | 0.710 (0.172) | 0.765 (0.157) | 0.803 (0.086) | 0.830 (0.066) |
| FT best-n (real) | 0.639 (0.149) | 0.684 (0.225) | 0.723 (0.173) | 0.780 (0.176) | 0.750 (0.178) | 0.830 (0.066) |
| FT best-n (RCA) | 0.639 (0.149) | 0.631 (0.234) | 0.687 (0.191) | 0.753 (0.166) | 0.722 (0.229) | 0.830 (0.066) |
| PL best-n (real) | 0.639 (0.149) | 0.625 (0.162) | 0.639 (0.123) | 0.640 (0.131) | 0.589 (0.196) | 0.553 (0.145) |
| PL best-n (RCA) | 0.639 (0.149) | 0.614 (0.146) | 0.640 (0.125) | 0.619 (0.580) | 0.632 (0.139) | 0.553 (0.145) |
| S+T best-n (real) | 0.639 (0.149) | 0.692 (0.164) | 0.747 (0.152) | 0.763 (0.151) | 0.786 (0.282) | 0.831 (0.063) |
| S+T best-n (RCA) | 0.639 (0.149) | 0.711 (0.160) | 0.755 (0.148) | 0.776 (0.282) | 0.797 (0.277) | 0.831 (0.063) |

Table 3: Different $n$-subject selection size in fine-tuning (FT), fine-tuning with pseudo-labels (PL), and training from scratch (S+T)

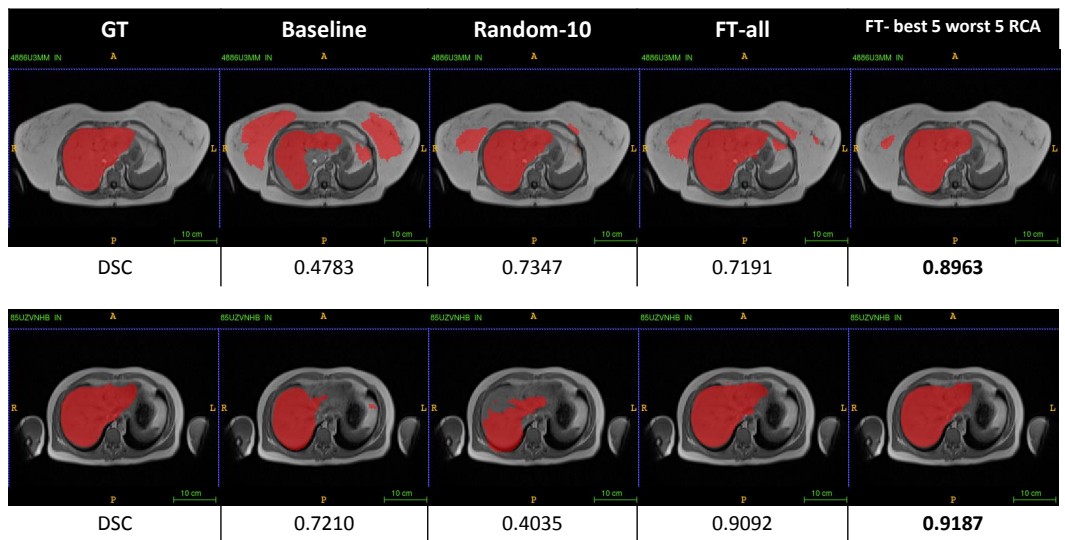

Figure 4: DARCA fine-tuning in liver segmentation. Combination of lowest and highest RCA prediction can give a better result than fine-tuning with random selection and with all of the target subjects.

### 3.4 Fine-tuning in Right Kidney Segmentation

From the three different strategies of DARCA in liver segmentation, we can see that fine-tuning with DARCA gives better results, less time-consuming (compared to training from scratch), and no iterative scheme needed. Also, from the results in liver segmentation (Table 2), combination of best-5 and worst-5 subjects always gives better or similar results than using all of the subjects from domain T, in all strategies. To validate these results, we also explore DARCA-FT in a different task: right kidney segmentation.

Similarly, the best results of right kidney segmentation with fine-tuning are achieved when we combine best-5 and worst-5 subject selection (see Table 4). The result (DSC: 0.716 with RCA selection) is better than when fine-tuning with all of the subjects from T (DSC: 0.658), and similar to when we train from scratch using all target subjects. This means we could cut the processing and annotation time. Figure 5 depicts some examples on how DARCA-FT with combination of best-5 and worst-5 subject selection improves the baseline and gives the best segmentation accuracies.

## 4  Discussion and Conclusion

Set size and subject selection are important in domain adaptation, where usually labels are not available in one of the domains. Thus, we explored whether it will be useful to select only the "valuable" subjects by RCA to be annotated. Pseudo-labels, which normally are used in semi-supervised learning and regularization, seem not to be useful in *supervised* DARCA. As observed in Figure 3 the performance drops as we increase the number of pseudo-labels in fine-tuning. Pseudo-labels will introduce more noise confusing the training of the networks and make them incapable to be applied to the new domain.

All of our strategies in DARCA (training from scratch, fine-tuning, and iterative) show a consistent result, combination of best-5 and worst-5 subject selection yields best results. RCA selection of those combined subjects also results in a similar accuracy to the real selection, compared to a bigger gap between RCA and real selection when fine-tuning with only best or worst subject selection.

In this scheme, DARCA shows its potential to leverage the highest and lowest confident subjects, to be incorporated in the domain adaptation process. We demonstrated that DARCA with few labeled data can perform similarly and/or better to full-size labeled target data. In the examples of Figure 4 and Figure 5, DARCA with best-5 and worst-5 subjects show consistent results across different tasks (liver and kidney segmentation).

In the case of real DSCs between (0.6, 0.8), RCA underestimates the predictions [12]. This led to a different subject selection. In future, an improvement of RCA prediction for medium level DSCs (0.6-0.8) needs to be investigated so that it can work more accurately. Our study only focuses at a predefined number of selected subjects, and a more thorough exploration needs to be done in future work. Traditional active learning may have more flexibility regarding the number of "valuable" samples to be included, but it requires iterations, which is time consuming. DARCA only needs RCA to predict the DSCs from the baseline segmentation where we can select the highest and lowest predicted subjects to be incorporated in a quick fine-tuning procedure. Therefore, we can conclude that DARCA could save processing time (no iterations needed) and annotation time with promising results avoiding the need for a large annotated target database.

**Acknowledgments**

V. Valindria is supported by the Indonesia Endowment for Education (LPDP)- Indonesian Presidential PhD Scholarship programme. K. Kamnitsas is supported by the Imperial College President's PhD Scholarship programme. B. Glocker received funding from the European Research Council (ERC) under the European Union's Horizon 2020 research and innovation programme (grant agreement No 757173, project MIRA, ERC-2017-STG).

The MRI data has been collected as part of the MALIBO project funded by the Efficacy and Mechanism Evaluation (EME) Programme, an MRC and NIHR partnership (EME project 13/122/01). The views expressed in this publication are those of the author(s) and not necessarily those of the MRC, NHS, NIHR or the Department of Health. This research has been conducted using the UK Biobank Resource under Application Number 12579.

| Strategies | Fine tuning (mean (stdv)) |
| --- | --- |
| Lower bound | 0.417 (0.263) |
| Training from scratch (all T) | 0.719 (0.106) |
| FT all T | 0.658 (0.114) |
| FT Random 5 | 0.506 (0.278) |
| FT worst 5 (real) | 0.416 (0.254) |
| FT worst 5 (RCA) | 0.358 (0.274) |
| FT best 5 (real) | 0.500 (0.293) |
| FT best 5( RCA) | 0.421 (0.319) |
| Best 5 and worst 5 (Real) | **0.726** (0.126) |
| Best 5 and worst 5 (RCA) | 0.716 (0.122) |

Table 4: DARCA-FT on right kidney segmentation. Combination of best and worst subject selection gives the best result, and RCA selection also gives a similar accuracy to the real selection.

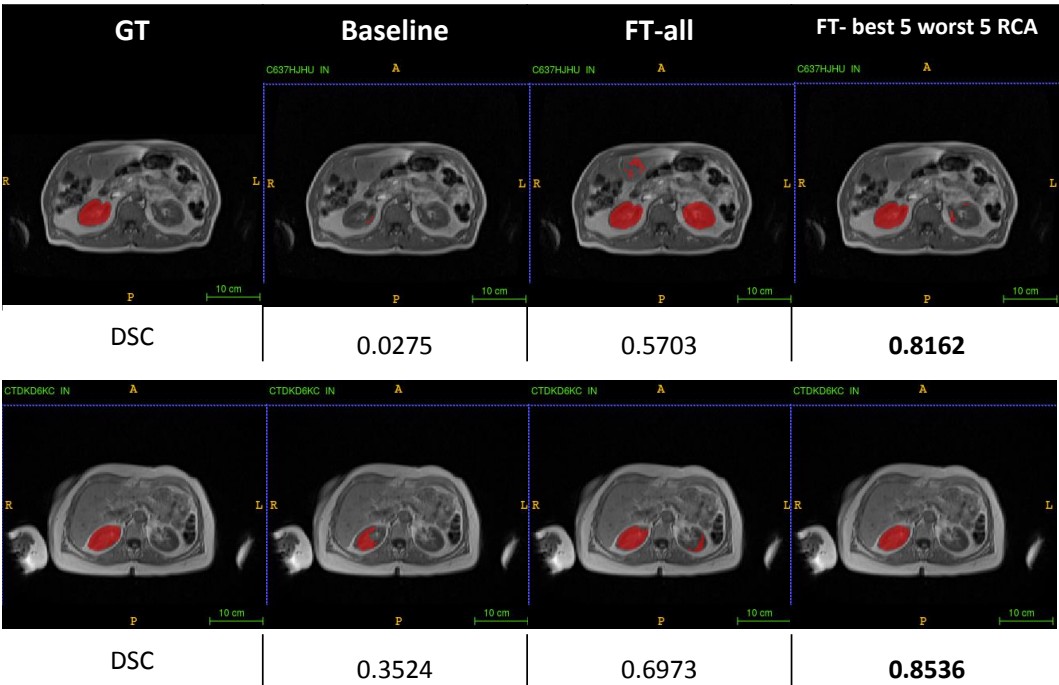

Figure 5: DARCA fine-tuning in right kidney segmentation. Combination of lowest and highest RCA prediction can give a better result than fine-tuning with all of the target subjects.

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
