# OpenReview forum: "Domain Adaptation for MRI Organ Segmentation using Reverse Classification Accuracy"
_MIDL.amsterdam/2018/Conference — MIDL 2018 Poster_

### Review · AnonReviewer1 · 2018-05-04
**important problem of leveraging knowledge from one domain to generalize to another domain, results are not fully compelling**

**Rating:** 3
**Confidence:** 2

**Review:**

The paper proposes a model that is trained on data from one scan and that automatically selects training samples from a second scan to improve the performance on the second scan's data. The method is tested to perform liver and kidney segmentation.

pros
+ important problem of leveraging knowledge from one domain (data from scan 1) to generalize to a second domain (data from scan 2)
+ simple method that could be used out of the box

cons
- results are not compelling, some choices seem arbitrary
- clarity could be improved

It is unclear whether the task presented in the paper is domain adaptation. Although the data is gathered from two different machines, and thus input distributions change across domains, it seems the problem is tackled from a transfer learning perspective (fine-tuning pre-trained models with limited amount of samples) or by training models from scratch (by merging data from the two domains). Could the authors address this question?

The abstract and introduction of the paper suggest the the paper will do organ segmentation, but reported results focus mainly on liver segmentation.

RCA is not properly explained in the paper; it would be beneficial to review the method in more details for the paper to be self-contained.

Table 1 should also report results obtained by the baseline when tested on a separate set of source data, to highlight the performance drop between the 2 domains. Does the baseline trained on T correspond to a model only trained on target data? The caption of the table should be self-contained.

It would be necessary to explain how the train/valid/test splits are performed and used in the experiments.

In general, notation in tables should be more improved; it is not straightforward to understand which results correspond to which methods, given the chosen names.

How do you pick the value of n=5? (it seems an arbitrary choice)

What is the reasoning behind picking subjects with high RCA score? It seems that picking sample with lower RCA score would be more valuable. Results on Table 2 suggest that picking the worst samples leads to better results. Why not pick an increasing number of bad samples instead of picking the 5 best and the 5 worst ones?

It would be worth highlighting that the so-called real methods are included for the sake of comparison with RCA performance.

Table 3 is referred to as Table 3.3. and there is no Table 3.2.

Table 3 contains results for increasing number of target samples. Which strategy is used to include samples (best/worst/random)? It would be interesting to compare how different strategies perform w.r.t. each other when increasing the number of subjects to be included for training. How do those strategies relate to curriculum learning?

Table 3 is too small.

Have the authors tried to fine-tune all the layers of the network instead of only the top ones?

Please improve the clarity of Figure 2.

In the pseudo-labels, do you use all the pseudo-labels as ground truth or set a threshold on the probabilities to decide which ones to use? If a threshold is applied, how is the method extended to handle dense predictions? Do you only compute error in some positions?

In the iterative DARCA, it is not clear whether the process can pick the same samples twice (at different iterations).

Please try to be consistent when using the model name, it seems that DARCA is introduced but barely used in the text, leading to confusion.

Figures 3 and 4 should be further discussed. Could the authors provide some intuitions on the performance drops observed in Figure 3?

Tables 2, 3 and 4 should also report std across folds.

Are the datasets used publicly available? Comparing to other models that appear in the medical imaging literature would be extremely beneficial to better assess the impact of the contribution of the paper.


**Special Issue:**

No

---

> ### Comment · ~Vanya_V_Valindria1 · 2018-05-15
> **Details and clarification**
>
> We thank the reviewer for the constructive feedback.
> - The task is domain adaptation since the two distribution considered as different domains. But we focus more to approach via the traditional transfer learning methods, as these approaches are commonly used in practice. Thus, we hope it will add benefit practically.
> - We will add more explanation about the RCA on the paper.
> - We will clarify the details/caption in tables to be more self-contained and explain how the data split for the experiment.
> - The value (n=5) was picked as we considered as it is a minimum (we have tried also with n = 2, 8, etc) number
> that can give an effect to DARCA, w.r.t to the whole size of testing data (T) (n = 45).
> - We have tried picking up the worst sample in increasing number, but the result is worse then the 'best 5 and worst 5'.
> - Yes, we have tried to fine-tuned all the layers of the network instead of only the top ones. The results are shown to be lower and more training time needed.
> - In pseudo-labels section, we simply used all of the segmentation results of test data from the baseline as the ground-truth.
> - In the iterative DARCA, it is not overlapped, as the process cannot pick the same samples twice at different iteration.
> - Unfortunately, the datasets are not publicly available.
> - We have taken all minor corrections into account and are thankful for pointers for future work.

---

### Review · AnonReviewer3 · 2018-05-09
**A domain adaptation framework . The use of RCA for subject selection is interesting and the different strategies are well explained.**

**Rating:** 4
**Confidence:** 2

**Review:**

The authors present a domain adaptation framework which is based on reverse classification accuracy (RCA) for subject selection for additional training/fine-tuning. Different strategies were compared as well as several variations in training size for two different segmentation problems. The presented problem is of interest and affects many medical imaging applications where the training sets varies from the test set.

This work if of interest for the MIDL conference community.  It deals with a known problem where the network model seems to perform well on a dataset from one hospital but performs poorly on a dataset taken from a different hospital. The use of RCA for subject selection is interesting and the different strategies are well explained.

Pros:
-	Novel approach using RCA for subject selection
-	Different strategies are explored
Cons:
-	The use of pseudo-labels seem to be incomplete. Different works showed that pseudo-labels can improve the network performance. I believe that this path should be explored more thoroughly.


**Special Issue:**

Yes

---

> ### Comment · ~Vanya_V_Valindria1 · 2018-05-15
> **Pseudo-labels in DARCA**
>
> Thank you for your comments and interest. Regarding the use of pseudo-labels, other networks used pseudo-labels in semi-supervised learning to leverage the unlabeled data. Meanwhile, our DARCA is more applied to supervised learning. These pesudo-labels can be considered as noisy labels in our experiment, as they produced the baseline prediction as of high quality to begin with, in which in domain adaptation it should be sub-optimal. As we do not consider it as a core contribution, we haven't evaluated pseudo-labels effect more thoroughly.

---

### Review · AnonReviewer2 · 2018-05-10
**Interesting concept and method, application is not compelling**

**Rating:** 4
**Confidence:** 2

**Review:**

The paper presents a method for updating a model trained against data from a specific platform or following a specific sequence to be updated to perform well against a new data set obtained using a different platform or sequence.  The method has two components first predicting the most important training samples for use in updating the model (by virtue of predicting the baseline model error on the new data) and then second the best way to update the model.  They evaluate the method on lung and kidney segmentation in two different whole body MRI datasets.

The introduction of the technique for predicting the baseline error is interesting and well described.  It would be interesting if the authors provided more details concerning the accuracy of the predictions.  The prediction accuracy is only mentioned briefly in passing, but as the final results seemed to show that including the samples with the largest and smallest predicted errors was important it would be useful to better understand the techniques ability.

The largest weakness of the paper was the choice of application.  Organ segmentation isn't a very useful application and in the future it would be interesting to see this technique applied to something more complex.   Lung nodule segmentation moving from thin slices to thick for example.

**Special Issue:**

No

---

> ### Comment · ~Vanya_V_Valindria1 · 2018-05-15
> **Accuracy prediction and application**
>
> Thank you for your response. We will improve the explanation and details about the accuracy prediction (RCA) method on the paper.
> Regarding the application of DARCA on organ segmentation, this paper shows that the approach can work on a common segmentation task, which can be useful to apply the approach further on other more complex tasks, such as lesion or pathology segmentation.

---

### Decision · Program_Chairs · 2018-05-15
**Paper28 Acceptance Decision**

Poster